# The Application of Micro-Vibratory Phenomena of a Shape-Memory Alloy Wire to a Novel Vibrator

**Takashi Chujo [1] and Hideyuki Sawada [2],\***

1 Department of Applied Physics, School of Advanced Science and Engineering, Waseda University, 3-4-1 Okubo, Shinjuku-ku, Tokyo 169-8555, Japan; chujo@akane.waseda.jp
2 Faculty of Science and Engineering, Waseda University, 3-4-1 Okubo, Shinjuku-ku, Tokyo 169-8555, Japan
\* Correspondence: sawada@waseda.jp

**Abstract:** The widespread use of smartphones and smart wearable devices has created a great demand for vibrators with complex vibration patterns driven by simple circuits. In our previous studies, we observed that a filiform shape-memory alloy (SMA) wire will shrink and then return to its initial length, perfectly synchronizing with a given pulse current. Here, we developed a novel vibrator whose structure allows the micro-vibrations of an SMA wire to be amplified up to a recognizable level without directly touching the wire. The vibrator has the advantage of independently controlling its magnitude and frequency together with a simple driving circuit since it is directly driven by a frequency-modulated pulse current with a controlled duty ratio. We measured the power consumption and the acceleration generated by the vibrator. The results showed that the vibrator consumed only 4–77 milliwatts of power with a quick vibration response within 5 milliseconds, and the acceleration increased significantly in a duty ratio range of around 1%. Furthermore, user evaluations demonstrated that differences in the magnitude and frequency of the generated vibrations were sufficiently recognized when the vibrator was driven by different duty ratios and frequencies, and the vibrator provided various tactile and haptic sensations to users.

**Keywords:** shape-memory alloy; vibrator; vibrotactile display





## 1. Introduction

Vibrators are basic components of smartphones and game controllers and have been used to convey two-valued information such as alarms and notifications by switching their vibrations on and off. The recent growth in smart wearable devices, VR, and AR has led to a great demand for more complex vibrations with vibrators driven by simple electrical circuits.

In terms of creating mechanical vibrations, there are two conventional methods: one using electromagnetic forces and the other using the piezoelectric effect. Moreover, the method using electromagnetic forces is classified into two types according to the direction of the presented vibration: Rotary Electromagnetic Actuators (REAs) and Linear Electromagnetic Actuators (LEAs). REAs generate vibrations by rotating a DC motor. The most typical REA is an Eccentric Rotating Mass (ERM) actuator, which generates vibrations by rotating a weight with an unbalanced center of mass. A recent study using an ERM actuator in a drone showed that two types of vibration feedback, weak and strong (corresponding to the distance from the obstacles in the drone's travel direction), can help the operator become aware of the environment around the device [1]. However, ERM actuators are impractical because of their large size, heavy weight, slow response speed, and non-independence in vibration magnitude and frequency, both of which depend on rotational speed. In particular, this dependency limits the variations that the ERM actuator can present. Indeed, the vibration feedback presented in [1] was limited to only three types: no vibration, weak vibration, and strong vibration. On the other hand, LEAs consist of

electromagnets, magnetic materials, and elastic materials such as springs, generating linear vibration by applying an alternating current to the electromagnets to generate oscillating motion in a moving element. Although LEAs generally have a faster response time and are more flexible when controlling the frequency and amplitude of the vibration than ERM actuators, their problem lies in their narrow driving frequency range, residual vibration, and the necessity of using both electric and magnetic circuits [2,3]. Piezo actuators (PZAs) create vibrations with the piezoelectric effect, which is the effect of mechanical deformation through the application of voltage. The advantages of PZAs are their thinness, fast response time, low power consumption, and wide driving frequency range. However, the voltage required to drive them is rather high; therefore, a dedicated boost converter is required. An 80 V boost converter was developed for the use of PZAs in smartphones [4]. However, it is complicated, and even 80 V is not a sufficient driving voltage for the effective use of some PZAs.

SMAs are well known for remembering their original shapes. They are deformable via external forces in the martensite phase at lower temperatures and return to their preliminary remembered shape when heated to transition into the austenite phase [5,6]. SMA wires are conventionally employed for their comparatively slow actuation since the deformation of the wire occurs because of heat conduction. In contrast, we discovered that a filiform SMA wire will shrink up to 5% and return to its initial length, synchronizing perfectly with a given pulse current, and the deformation can be controlled with high frequencies of up to 1000 Hz. Unlike other microwires, such as amorphous wires [7,8], SMA wires instantly shrink longitudinally just by applying an electric current owing to the physical properties of their shape-memory effects. Hence, the components required to drive an SMA wire are simple and have the potential to be compact and lightweight. Additionally, shrinkage can be controlled via the duty ratio of the driving pulse. By utilizing the micro-vibratory phenomenon, we developed tactile displays that can directly create various tactile sensations on human skin [9–11].

In this paper, we propose a novel vibrator using the micro-vibrating effect of an SMA wire. This vibrator has the advantage of independently controlling the frequency and magnitude of its generated vibrations by using a simple driving circuit, which does not require a high driving voltage or a strong magnetic field. The vibrator generates sufficiently recognizable vibrations without directly touching micro-vibrating objects, which is different from our previously developed tactile displays. We developed a prototype vibrator using an SMA wire and measured the power consumption and acceleration corresponding to the force generated by the vibrator to investigate their characteristics regarding the driving pulse current. Furthermore, a vibration evaluation experiment was conducted with five subjects, in which the vibrator was driven by different duty ratios and input pulse frequencies. The evaluation results showed that it is possible to create a variety of vibrations using an SMA vibrator. This study contributes to the expansion of vibration creation methods.

## 2. Materials and Methods

### 2.1. Shape-Memory Alloy and Its Physical Properties

Shape-memory alloys (SMAs) are a class of shape-memory materials that can recover their original shape after quasi-plastic deformation [12]. The unique characteristics of SMAs are governed by a reversible solid phase transition between two phases: austenite and martensite. Austenite is a higher-temperature phase that has a predetermined shape, while martensite is a lower-temperature phase that is soft and easily deformable. The transitions from one phase to another begin and finish at critical temperatures $M_s$, $M_f$, $A_s$, and $A_f$, which depend on the composition of the alloy, its thermo-mechanical history, and the applied load.

Figure 1 presents a schematic diagram of the crystal structures of martensite and austenite for an SMA and the transformation between them. Upon cooling an SMA, the austenite phase collapses into a twinned martensite phase at the martensite starting

temperature ($M_s$) and completes the transformation at the martensite finishing temperature ($M_f$). Afterward, when stressed at a temperature below the austenite starting temperature ($A_s$), it transforms into detwinned martensite from twinned martensite and changes its shape. As the deformed SMA is heated, it starts to regain its original shape above $A_s$, and the transformation is completed at the austenite finishing temperature ($A_f$). Cooling the SMA below $M_f$ under no load results in a phase transformation from austenite to twinned martensite while also maintaining its shape. The above process is called the shape-memory effect (SME). In addition to thermally induced phase transformations such as the SME, the transformation from austenite to detwinned martensite can be induced by applying a sufficient mechanical load at a temperature above $M_s$. When just unloaded after the transformation, the SMA recovers from the deformed shape. This behavior is known as the pseudoelastic effect (PE). The shape partially recovers if heating is stopped at a temperature above $A_s$ but below $A_f$ in the SME process. This is because not all martensite transforms into austenite. Figure 2 illustrates the hysteretic and nonlinear relationship between temperature and the martensitic fraction of an SMA ($\xi$). At the temperature around the midpoint between the starting and finishing temperatures of the transformation, a significant change in the martensitic fraction occurs.

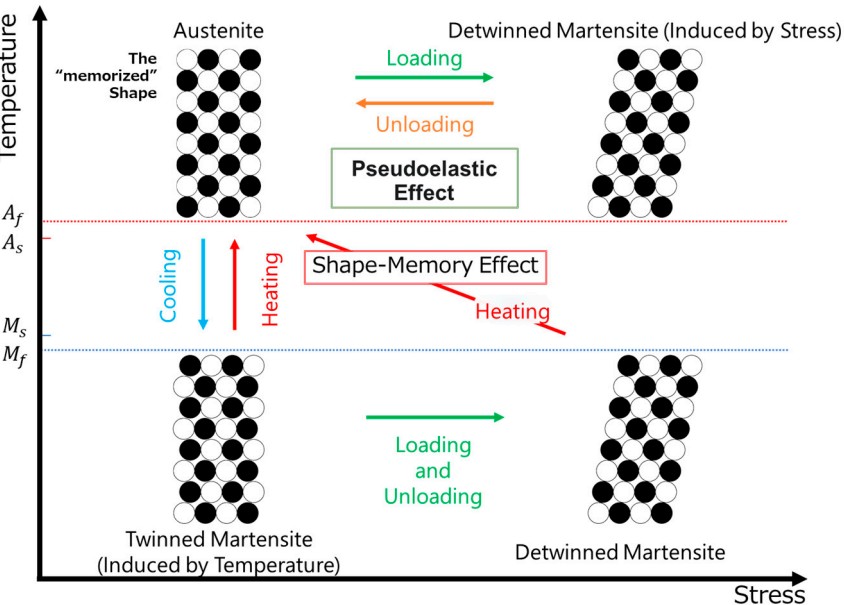

**Figure 1.** The SMA phase transformation model via stress application and temperature change.

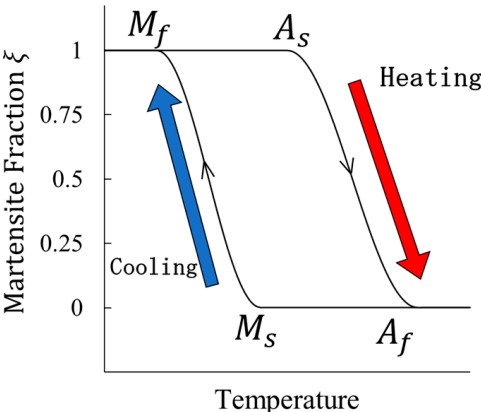

**Figure 2.** Conceptual diagram of the relationship between temperature and the martensite fraction. This graph was made by referring to the $\xi - T$ relationship expressed by cosine functions, as proposed by Liang and Rogers [13].

### 2.2. Micro-Vibration Actuator Using an SMA Wire

Filiform SMA wires have a unique characteristic in that they swiftly respond to temperature. When applying an electric current to an SMA wire, its temperature will rise because of Joule heating, and it will instantly shrink because of the phase transition from martensite to austenite. When the current stops, the wire returns to its original length by radiating heat. The small diameter leads to rapid heat dissipation because of the large surface area of the wire. The above process is repeated by applying continuous pulse currents to the SMA wire; as a result, it vibrates corresponding to the pulse. Pulse currents (see Figure 3) are characterized by three independent parameters: the amplitude, the frequency, and the duty ratio, which is the percentage of the pulse width to the period. Therefore, the vibration can be controlled by these parameters.

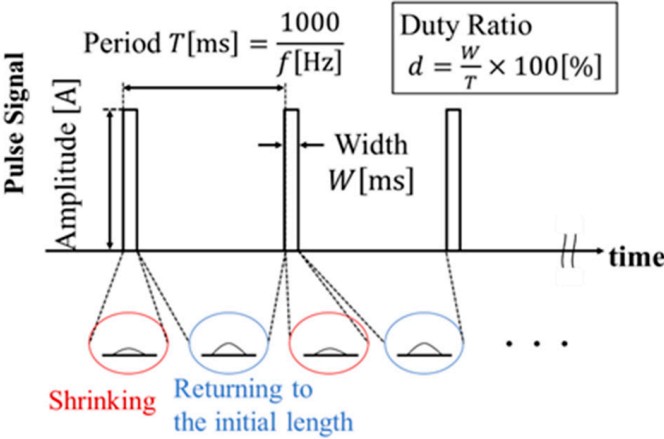

**Figure 3.** Continuous pulse current signal for driving an SMA wire.

### 2.3. Novel Structure of a Vibrator Using an SMA Wire

We developed a vibrator using one SMA wire as the motivating power source. To amplify the micro-vibration up to a sufficiently recognizable level for humans, we introduced a novel structure to the vibrator, as shown in Figure 4. The vibrator consists of five parts: an SMA wire, a brass pipe for heat radiation, a moving element with a weight to amplify the micro-vibration, a spring for tension application, and a base to secure other parts.

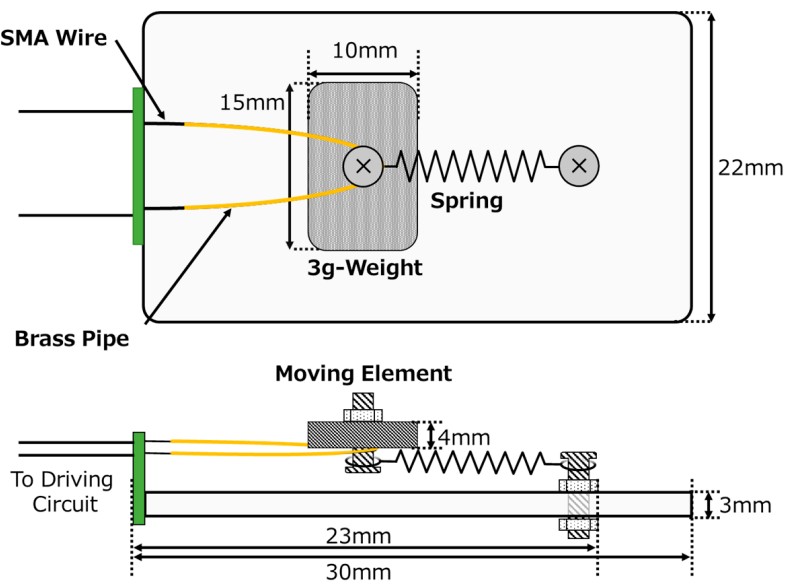

**Figure 4.** The structure of the SMA vibrator.

Figure 5 illustrates the working principle: the vibrator converts the micro-vibrations of an SMA wire into a recognizable force. In this study, continuous pulse currents were used to drive the SMA wire. In the non-driven status, stress is applied to the SMA wire by the spring to increase the response speed when the current flows and to prevent the moving element from freely moving in the OFF state. As the current passes, the wire is heated and shrinks instantly; therefore, the moving element shifts in the direction of the stretching spring. When the current turns off, the wire cools down and undergoes a martensitic transformation; thus, the moving element returns to its initial position because of the spring force. The brass pipe is used to effectively cool the SMA wire. By repeating the above process, the micro-vibrations of the SMA wire are amplified by the moving element–spring structure, and the amplified vibration created by the reaction force is directly conducted into the base part. One can feel the vibration by touching the base part of the vibrator, which can be efficiently applied to mobile devices such as smartphones and smartwatches.

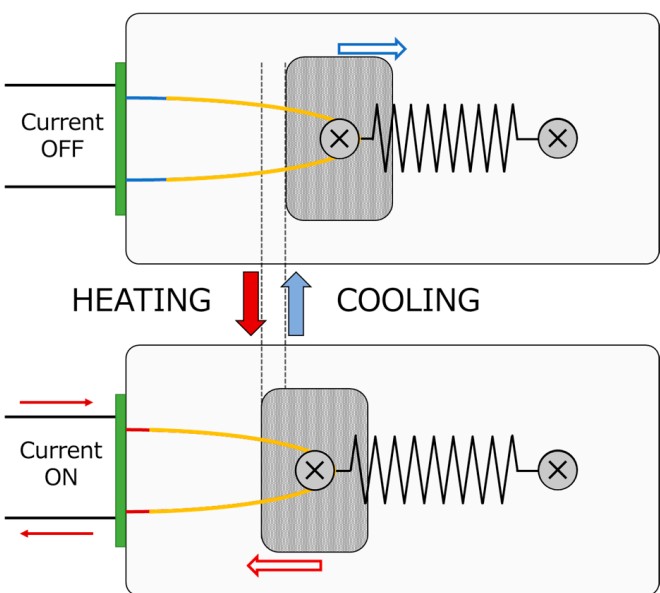

**Figure 5.** The principle of vibration generation.

Figure 6 shows two pictures of the prototype vibrator. The SMA wire is a BioMetal Fiber 100 (Toki Cooperation, Tokyo, Japan) with a length of 2 cm and is composed of NiTiCu. The physical properties of the wire are shown in Table 1. The austenite starting temperature ($A_s$) is about 70 °C, and the martensite starting temperature ($M_s$) is about 65 °C [14]. Both ends of the SMA wire are wrapped around copper eyelets and soldered to a 1 cm square universal board because SMAs do not have solderability to copper. The spring is a Φ 0.3 mm SUS304 tension spring with a coil diameter of 3 mm, a length of 10 mm, and a spring constant of 0.2 N/mm. The weight is a 3 g block comprising a TAMIYA multi-setting weight with a volume of $15 \times 10 \times 4$ mm$^3$. The base part is an acrylic plate of $30 \times 22 \times 3$ mm$^3$.

A circuit diagram of the control circuit used to drive the vibrator is shown in Figure 7. The transistor is an NPN epitaxial transistor 2SD880L from Unisonic Technologies Co., Ltd. (New Taipei City, Taiwan), and the DC power supply is a USB power adapter (W010A051) from Apple Inc. (Cupertino, CA, USA) with a rated voltage of 5.1 V and a rated current of 2.1 A. A continuous pulse voltage signal, whose frequency and duty ratio are programmed and output by an Arduino UNO, is applied to the transistor's base. The amplitude of the pulse voltage signal is 5 V, which saturates the transistor when the signal is turned on. Therefore, with the switching action of the transistor, the frequency and duty ratio of the pulse current sent through the SMA are controlled by the pulse voltage using the Arduino UNO.

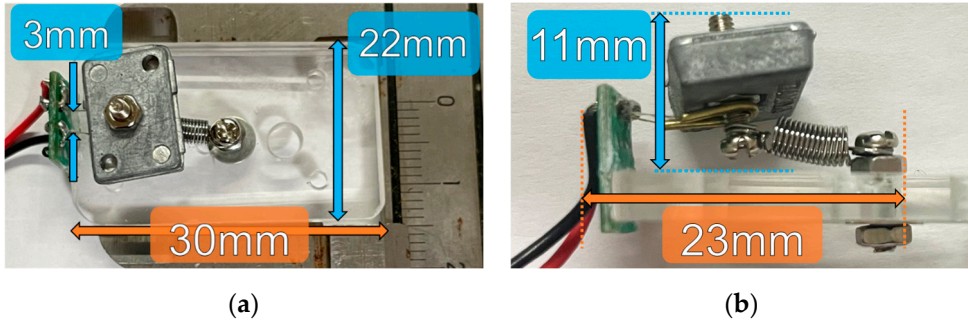

(**a**) (**b**)

**Figure 6.** The appearance of the prototype vibrator: (**a**) The vibrator viewed from above, the size of the base, and the width of the SMA wire. (**b**) The vibrator viewed from the side and the width and height of the fixed part.

**Table 1.** Physical properties of BioMetal Fiber 100.

| Physical Property | Value |
| --- | --- |
| Standard diameter (μm) | 100 |
| Practical force produced (load) (gf) | 70 |
| Practical kinetic strain (%) | 4.0 |
| Standard drive current (mA) | 200 |
| Standard drive voltage (V/m) | 27 |
| Standard power (W/m) | 5.40 |
| Standard resistance (Ω/m) | 135 |
| Tensile strength (Kgf) | 0.8 |
| Weight (mg/m) | 50 |

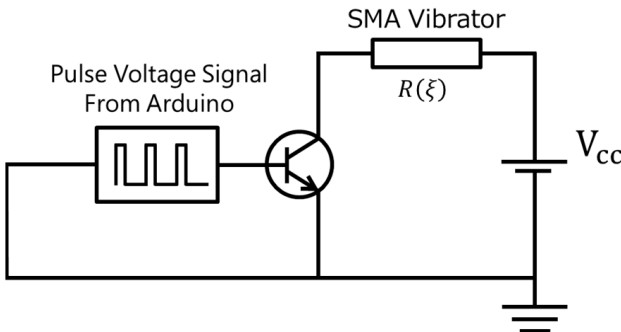

**Figure 7.** Control circuit for an SMA wire actuator. $R(\xi)$ is the electrical resistance of the SMA depending on the martensite fraction, $\xi$.

## 3. Results

### 3.1. Evaluation of Vibration Performance

3.1.1. Experimental Setup for Measuring Applied Electrical Power and Generated Acceleration

To investigate how the characteristics of the SMA wire affect the output of the vibrator when driving it at frequencies lower than the resonant frequency of its spring–mass system, a voltage and a current were applied to the vibrator and examined together with the generated acceleration. The electrical quantities were measured by connecting an INA219 sensor module to the power supply in series with the SMA wire. A schematic diagram of the acceleration measurement system is shown in Figure 8, and its appearance is shown in Figure 9. Coupled with a three-axis acceleration sensor (KXR94-2050 module), the vibrator was suspended by thin threads to allow it to freely move via vibrations. An aluminum frame fixture (weight, 787 g) was set up on a 5 mm thick rubber mat to absorb external vibrations. Two cotton threads were tied to the fixtures, and the lengths of the threads were adjusted to 15 cm. An accelerometer was fixed to an acrylic plate, which was connected

to the base of the SMA vibrator by four spacers. Two threads were placed inside the two acrylic plates and the spacers. The total mass of the vibrators, accelerometers, and spacers was 15.8 g.

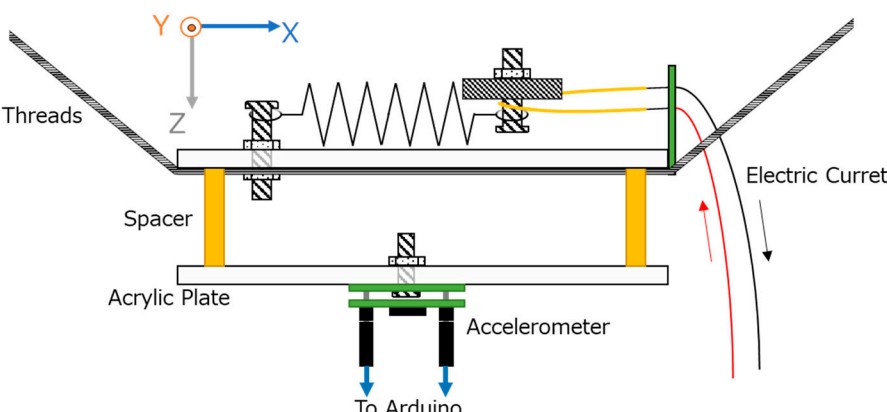

**Figure 8.** Schematic diagram of acceleration measurement system.

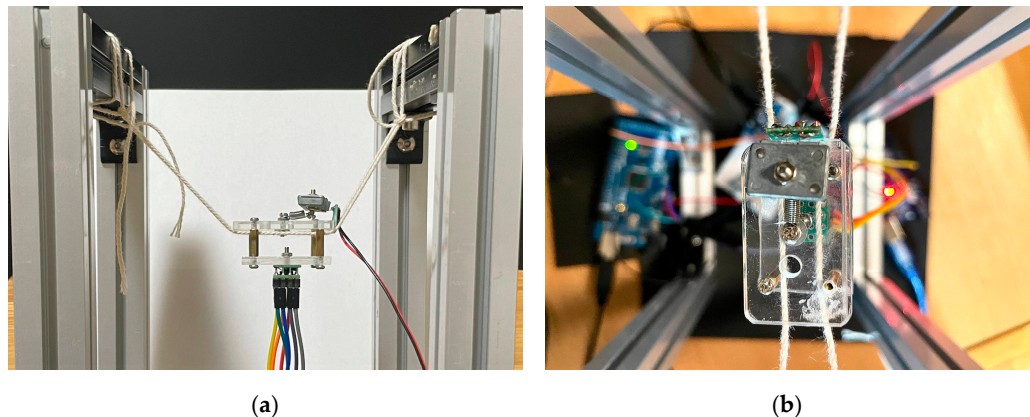

| (a) | (b) |

**Figure 9.** Photographs of the acceleration measurement system: (**a**) front view; (**b**) top view.

The measured data set consists of the elapsed time from the start of the measurement; the voltage and current applied to the SMA wire; and the generated acceleration in the X, Y, and Z directions (shown in Figure 8), which were recorded using an Arduino Mega. The measured data were sent to a PC with a sampling interval of 5 milliseconds by using DataStreamer, which enabled a data transfer to Microsoft® Excel® for Microsoft 365 MSO (Version 2212) in real time. The measurements were taken under 110 conditions: driving pulses with frequencies of 5, 10, 20, 30, 40, 50, 60, 70, 80, and 90 Hz and duty ratios varying from 0.5% to 1.5% in steps of 0.1% at every frequency. A total of 3000 sets of data were recorded for each condition.

### 3.1.2. Power Consumption Measurement

The effective power consumption was calculated as $100d \times \overline{VI}$ based on the product of the duty ratio and the average values of the voltage and current when the current flows. Figure 10 illustrates the relationship between the duty ratio and the estimated power consumption for each frequency. Table 2 shows the coefficients of determination, slope, and intercept of the approximate line obtained via linear regression, together with their respective standard errors. The distributions in Figure 10 and the coefficients of determination ($R^2$) are close to 1, demonstrating that the electrical power consumption linearly increased after raising the duty ratio without changing the driving frequency, as shown in Table 2.

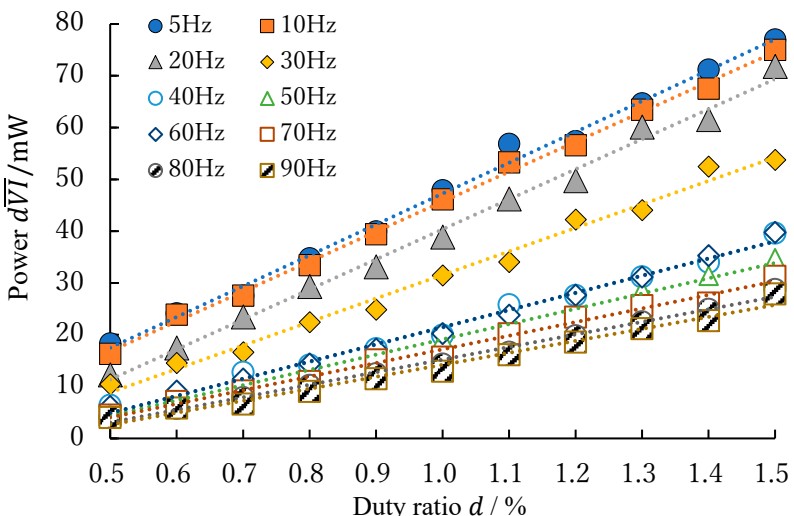

**Figure 10.** Relationship between duty ratio and power at each frequency.

**Table 2.** Regression analysis with duty ratio as the explanatory variable and electric power as the objective variable.

| Frequency /Hz | $R^2$ / | Slope /mW/% | Standard Error /mW/% | Intercept /mW | Standard Error /mW |
|---|---|---|---|---|---|
| 5 | 0.994 | 59.5 | 1.56 | −12.3 | 1.63 |
| 10 | 0.998 | 57.8 | 0.96 | −12.1 | 1.01 |
| 20 | 0.993 | 58.0 | 1.63 | −17.6 | 1.71 |
| 30 | 0.989 | 45.4 | 1.62 | −13.8 | 1.70 |
| 40 | 0.989 | 33.2 | 1.16 | −11.8 | 1.22 |
| 50 | 0.990 | 29.5 | 0.98 | −10.3 | 1.02 |
| 60 | 0.992 | 33.2 | 0.99 | −11.7 | 1.04 |
| 70 | 0.989 | 26.3 | 0.93 | −9.1 | 0.98 |
| 80 | 0.992 | 24.3 | 0.71 | −9.0 | 0.74 |
| 90 | 0.983 | 23.1 | 1.02 | −8.9 | 1.07 |

### 3.1.3. Acceleration Measurement

To remove the effect of gravitational acceleration from the measured acceleration, the acceleration $a$ generated by the vibrator was calculated as the vector $a_m - a_0$: the initial acceleration $a_0$ subtracted from the measured acceleration $a_m$, which is set as a vector consisting of the mode values of the acceleration in each axis direction, measured before driving the vibrator. In addition, we defined the maximum value of the acceleration in the $n$th period as $a_{\max,n}(f,d)$, which is the acceleration generated in one period.

Figure 11 shows the experimental results, summarizing the relationship between the duty ratio $d$ and the average acceleration in each pulse current condition, $\overline{a_{\max}}(f,d)$. Table 3 shows the coefficients obtained via linear regression between $\overline{a_{\max}}(f,d)$ and $d$. As shown in Figure 11, the relationship between $\overline{a_{\max}}(f,d)$ and $d$ shows that the generated acceleration does not increase linearly, as the duty ratio rises with each vibration frequency. Furthermore, at 5 Hz and 10 Hz, the acceleration increases significantly within a duty ratio ranging from 0.8% to 1.2%.

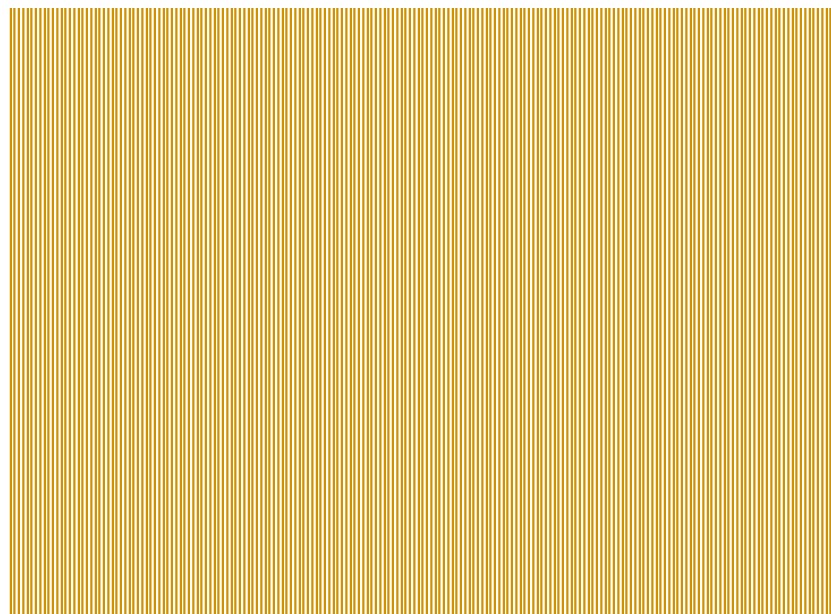

**Figure 11.** Relationship between the duty ratio at every frequency and the average magnitude of acceleration: $\overline{a_{\max}}(f,d)$.

**Table 3.** Regression analysis with duty ratio as the explanatory variable and $\overline{a_{\max}}(f,d)$ as the objective variable.

| Frequency /Hz | $R^2$ / | Slope /G/% | Standard Error /G/% | Intercept /G | Standard Error /G |
|---|---|---|---|---|---|
| 5 | 0.883 | 0.17 | 0.021 | $-0.04$ | 0.022 |
| 10 | 0.879 | 0.11 | 0.014 | $-0.01$ | 0.015 |
| 20 | 0.887 | 0.03 | 0.004 | 0.05 | 0.004 |
| 30 | 0.856 | 0.01 | 0.002 | 0.03 | 0.002 |
| 40 | 0.960 | 0.01 | 0.001 | 0.02 | 0.001 |
| 50 | 0.955 | 0.04 | 0.003 | 0.00 | 0.003 |
| 60 | 0.708 | 0.01 | 0.003 | 0.03 | 0.003 |
| 70 | 0.922 | 0.04 | 0.004 | 0.00 | 0.004 |
| 80 | 0.802 | 0.03 | 0.005 | 0.01 | 0.005 |
| 90 | 0.234 | 0.02 | 0.011 | 0.01 | 0.012 |

Figure 12 shows the time variation in acceleration for one cycle at $d = 1.3\%$ since the generated acceleration is significantly increased at this duty ratio. The measurement results indicate that the X-component of the acceleration is greater than in the other two components. Furthermore, the magnitude of acceleration reached its maximum at around 5 ms and decreased at around 10 ms for all frequencies.

### 3.2. Vibration Evaluation by Users

3.2.1. Experimental Conditions for Vibration Evaluation

Next, we conducted an experiment to verify how humans perceive vibrations generated by the vibrator when the duty ratio changes at a fixed frequency and the frequency changes at a fixed duty ratio. Five volunteers were provided vibratory stimuli created by the vibrator and provided feedback on the perceived vibrations without being informed of the parameters of the input pulse signal. All participants were Japanese and in their 20s, consisting of four males and one female, with standard sensitivities to vibratory stimuli.

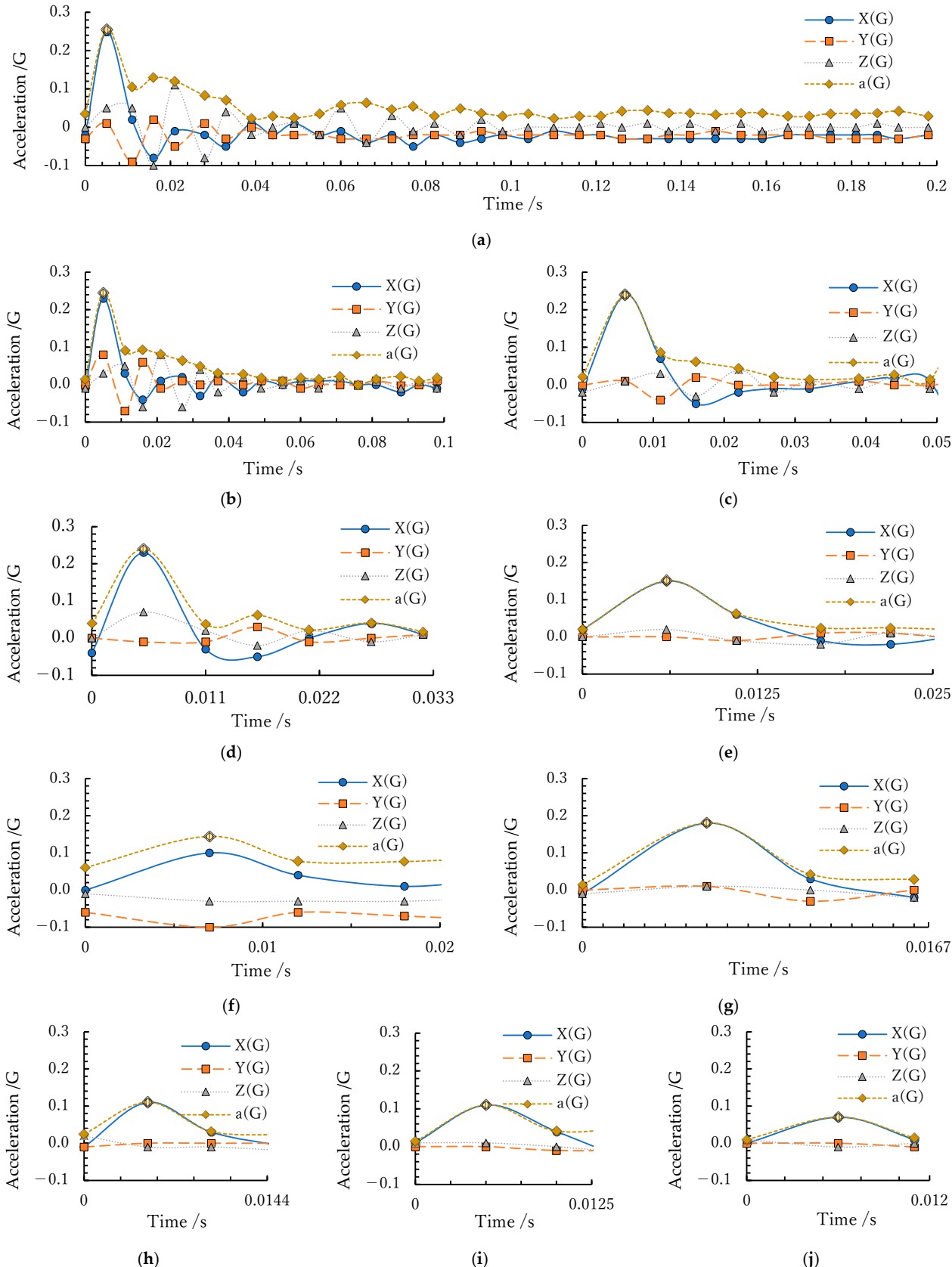

**Figure 12.** Time variation in acceleration for one cycle at $d = 1.3\%$, where the magnitude of vibration seems to be sufficiently saturated. The driving pulse used the following frequencies: (**a**) 5Hz, (**b**) 10Hz, (**c**) 20Hz, (**d**) 30Hz, (**e**) 40Hz, (**f**) 50Hz, (**g**) 60Hz, (**h**) 70Hz, (**i**) 80Hz, (**j**) 90Hz.

A. Changing Duty Ratio Conditions

The participants evaluated the magnitude of vibrations when the vibrator was driven by continuous pulse currents with five different duty ratios for each frequency (10, 50, and 100 Hz). Based on the results of the acceleration measurement, 1.0% was the middle of the duty ratio interval, during which the generated acceleration increased at low frequencies. When the vibrator was driven at a duty ratio of 1.0%, the participants pinched the short edges of the base, and this vibration magnitude was used as a level 5 reference magnitude for each frequency. Based on a comparison with the reference magnitude, the participants scored the perceived vibration magnitudes on a scale of 0 to 10 at duty ratios of 0.5, 0.7, 1.0, 1.2, and 1.5% when picking up the short edges of the base and then the long edges, except for when the reference magnitude was determined.

B. Changing Frequency Conditions

The participants were provided with vibrations driven by 1% duty ratio pulse signals with frequencies of 5, 10, 20, 40, 80, 120, 240, and 480 Hz. They provided feedback on how much they felt these eight vibration types and a relative evaluation of the magnitude of the perceived vibrations on a scale from 0 to 10.

3.2.2. Evaluation Experiment and the Results

Figure 13 shows the results of the evaluation of the perceived vibrations under the conditions introduced above. The bar graphs show the evaluation values of the vibration magnitudes for all five subjects, and the line with data markers shows the average values for the evaluations in different conditions. Under all conditions, the average value tended to increase as the driving duty ratio rose. Additionally, the perceived sensation became greater when a subject pinched the long sides, as compared with pinching the short sides. With 10 Hz pulses, the subjects could distinguish all or most of the vibrations generated by the vibrator driven at different duty ratios. The subjects reported that it was more difficult to distinguish vibrations when the vibrator was driven by higher-frequency pulses.

As for the eight frequencies within 5–480 Hz, the perceived relative magnitudes are shown in Figure 14. Table 4 shows the responses to the following question: "How do you feel the vibration from No.1 to No.8? e.g., simile for other sensation, onomatopoeia, fineness of the perceived vibration". Figure 14 indicates that the higher-frequency vibration generated at a 1% duty ratio was recognized as smaller, except for Subject 1 and Subject 4, who evaluated the magnitude of the 40 Hz vibration as larger. As shown in Table 4, the 5 Hz and 10 Hz vibrations created a sensation like pulsation or being poked, and the 10 Hz vibration was perceived faster. The 20 Hz vibration caused a trembling sensation. Regarding the 40 Hz and higher-frequency vibration, the subjects reported that it was difficult to distinguish each pulse, and the higher-frequency vibration was felt as a finer vibration. Subject 5 reported that the 240 Hz vibration was imperceptible. As for the 480 Hz vibration, none of the subjects perceived it as a tactile vibration.

**Table 4.** Correspondence between the label number and the frequency and subjective evaluation of the perceived vibrations generated by the SMA vibrator driven by 1% duty ratio pulse currents with 8 kinds of frequencies.

| Number | Frequency/Hz | Subject 1 | Subject 2 | Subject 3 | Subject 4 | Subject 5 |
|---|---|---|---|---|---|---|
| 1 | 5 | Heavy vibrations; poked or slapped sensation with a thin rod | Tap-tap, like a little fast pulsation | Like pulsating | Coarse and poke-like sensation | Like a heartbeat |
| 2 | 10 | Busy poking; like weak machine-gun stimulation | To-to-to-to, like a very fast pulse | Felt like really getting out of breath | Fine sensation compared with No.1 | Felt like pouring liquid |
| 3 | 20 | Colicky sensations | Raspy | Just barely coarse | A sensation of trembling in small increments | Felt like holding an insect |
| 4 | 40 | Rough feeling; most stimulating | Susurrus, like a numb feeling | Fast | A sensation of numbing rather than poking; harder to distinguish each pulse than Nos. 1 to 3 | Felt like the wings of a cicada vibrating |

**Table 4.** *Cont.*

| Number | Frequency/Hz | Subject 1 | Subject 2 | Subject 3 | Subject 4 | Subject 5 |
|---|---|---|---|---|---|---|
| 5 | 80 | Fine vibrations; like a fluffy sensation | Vroom Zzzz | Fine, but felt the "grain" of vibration | Numbing sensation | Felt slightly ticklish |
| 6 | 120 | Finer or fluffier vibration than No.5; like grabbing something soft | Voom | Very fine | Little or no sensation of vibration | Fine vibrations |
| 7 | 240 | Very fine vibration; did not feel much | The sensation of tracing things at a distance that may or may not touch | Smooth | Did not feel as if it was vibrating | Did not feel the vibration |
| 8 | 480 | No vibration was felt, but a sound was heard | Felt nothing | Little vibration felt | Unable to perceive the vibrations | Did not feel the vibration |

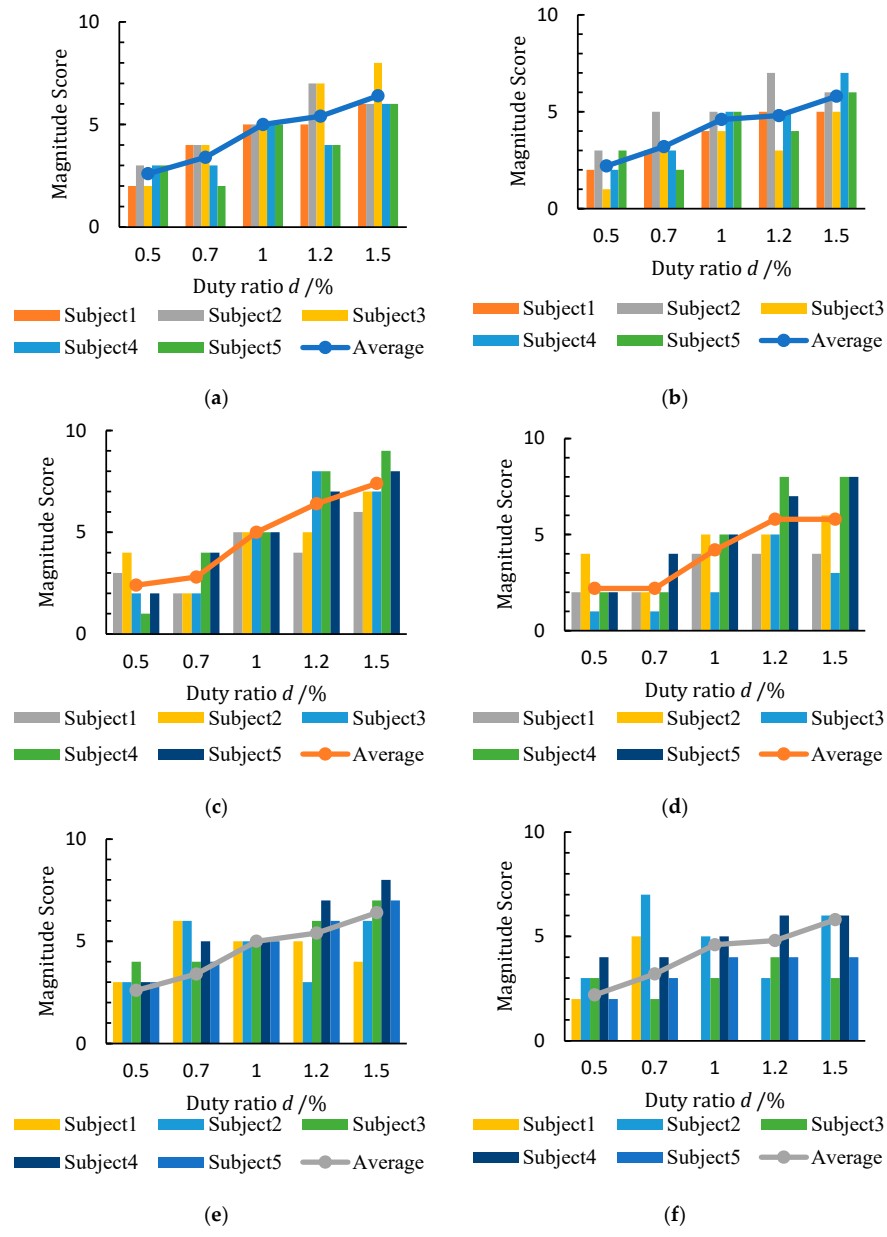

**Figure 13.** Magnitude evaluation of the perceived vibrations generated by the vibrator, driven by pulse currents with 0.5–1.5% duty ratios and 10, 50, and 100 Hz frequencies: (**a**) 10 Hz, picking up the short edges; (**b**) 10 Hz, picking up the long edges; (**c**) 50 Hz, picking up the short edges; (**d**) 50 Hz, picking up the long edges; (**e**) 100 Hz, picking up the short edges; (**f**) 100 Hz, picking up the long edges.

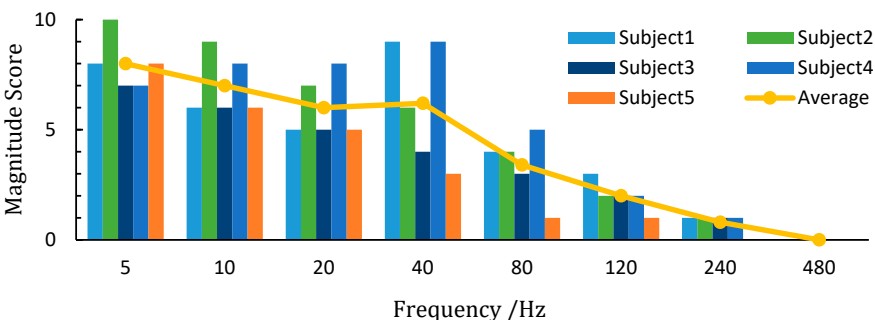

**Figure 14.** Magnitude evaluation of the perceived vibrations generated by the vibrator, driven by 1% duty ratios and 5–480 Hz frequencies.

## 4. Discussion

We proposed a novel vibrator that generated sufficient vibrations by utilizing the micro-vibrations of an SMA wire and a mass–spring structure. The vibrator was driven by controlling the frequency and duty ratio of the electric current sent through the SMA wire. The voltage and the energy applied to the wire were about 5 V and 4–77 mW, generating micro-vibrations with controlled frequencies. Using driving voltage is advantageous compared with piezoelectric actuators, which require around 100 V. Furthermore, unlike electromagnetic actuators, no magnetic field is required, and neither electromagnets nor permanent magnets are used. Therefore, there is no need to consider the effects of the magnetic field secondarily generated by the vibrator on other components. The result of the acceleration measurement showed that the generated acceleration significantly increased as the duty ratio rose without changing the driving frequency. Moreover, the score of the vibration evaluation further showed that the vibration intensity could be changed by controlling the duty ratio. This is advantageous compared with ERM actuators, whose frequency and vibration amplitude are not independently controlled.

As shown by the results of the acceleration measurement and the vibration evaluation, the X-directional acceleration—that is, the magnitude of the perceived vibration when the short edge of the vibrator was pinched—was greater. This is because the magnitude of the vibrations generated along the direction the moving element moves is greater. Furthermore, a larger vibration tends to be obtained at lower frequencies, although the resonant frequency of the spring–mass system (consisting of a tension spring and a moving element) is $4 \times 10^2$ Hz. Therefore, the nonlinearity between acceleration and the duty ratio is explained by the nonlinearity between the martensite transition and the temperature variation, rather than the resonance of the spring–mass system. In the duty ratio range where the acceleration increased significantly, the temperature at the end of the electric current application was seemingly around the middle of $A_s$ and $A_f$; as a result, the martensite fraction significantly decreased as the duty ratio increased, and the generated acceleration increased sharply. On the other hand, under conditions with lower duty ratios or a high frequency with short heating time, the SMA's temperature at the time of switching from ON to OFF seemed to be close to $A_s$; the martensite fraction barely changed because of the temperature variation. In addition, under the conditions of low frequency and a high duty ratio, the SMA's temperature was near $A_f$ or higher, which also led to a small change in the martensite fraction regarding temperature variation. The small changes in the martensite fraction caused few changes in the generated force, corresponding to the acceleration generated during duty ratio variations. The controllable range of vibration magnitude for higher frequencies will expand if the duty ratio increases in a range where the temperature is higher and when the driving pulse current switching from ON to OFF leads to a significant increase in generated acceleration, similar to those for low frequencies, as shown in Figure 10a,b.

The acceleration measurement results and the vibration evaluation indicated that the magnitude and frequency of generated vibration were independently changed by

controlling the duty ratio and the frequency of the driving pulse current; the vibrator was able to provide various vibratory stimuli to a user. For example, in [1], where vibration feedback with an ERM actuator was used to define a drone's surrounding environment, the distance from obstacles could only be expressed in three levels; by using the proposed SMA-based vibrator, the four directions (front, back, left, and right) can be expressed by four different frequencies, and the distance from an object can be represented by the magnitude of the generated vibration patterns.

The next challenge for this SMA-based vibrator is downsizing its body by optimizing its entire physical structure. Although the SMA wire itself is small and lightweight, the introduced structure, including the weighting block, the spring, and the base, made the vibrator large and heavy. In addition, a spring with a maximum spring constant was employed in the prototype device to obtain greater resonance frequency in the spring–mass system.

As with the SMA vibrator proposed in this study, voice coil actuators are generally used to present one-directional vibrations; however, in recent years, work has been conducted on developing multi-degree-of-freedom actuators using voice coils [15,16]. By increasing the number of SMA wires used to shift a movable mass, it should be possible to achieve multidirectional vibrations. Furthermore, our SMA wire actuator was driven by a continuous pulse current with a constant frequency and duty ratio. It is possible that more complex vibrations can be created using a vibrator by changing the shape of the electric current. If multidimensional and complex vibrations can be generated by using micro-vibrating SMA wires, SMA vibrators will become useful and sophisticated devices for haptic and vibrotactile displays.

## 5. Conclusions

In this study, we proposed a novel vibrator using the micro-vibratory phenomena of an SMA wire. It was driven by continuous pulse currents with an energy consumption rate of 4–77 mW using a simple switching circuit with a 5 V power source. Our measurements showed that the electrical power consumption linearly increased along with an increase in the duty ratio of the driving pulse, while the generated acceleration significantly increased with a duty ratio range of 0.8–1.2%. The mass–spring structure allowed the micro-vibrations of the SMA wire to be amplified up to a recognizable level. The results of the generated acceleration measurement and the vibration evaluation indicated that the magnitude and frequency of the generated vibrations could be changed independently by controlling the duty ratio and frequency of the driving pulse current; thus, the vibrator could create various vibratory stimuli. This study expands vibration-creating methods for handheld and wearable devices by proposing an SMA-based vibrator with different characteristics compared with other vibrotactile displays.

**Author Contributions:** Conceptualization, T.C. and H.S.; data curation, T.C.; formal analysis, T.C. and H.S.; funding acquisition, H.S.; investigation, T.C. and H.S.; methodology, T.C.; project administration, H.S.; resources, H.S.; software, T.C.; supervision, H.S.; validation, T.C. and H.S.; visualization, T.C.; writing—original draft, T.C.; writing—review and editing, H.S. All authors have read and agreed to the published version of the manuscript.

**Funding:** This research was supported by the JSPS KAKENHI Grant-in-Aid for Scientific Research (B) 20H04214 and the Hagiwara Foundation of Japan 3rd Research Grant.

**Data Availability Statement:** The data obtained by the user experiments are not included to maintain the privacy of the subjects. Other data are presented in the manuscript.

**Conflicts of Interest:** The authors declare no conflict of interest.

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
