# Peer review of "The Application of Micro-Vibratory Phenomena of a Shape-Memory Alloy Wire to a Novel Vibrator"

_vibration, doi:10.3390/vibration6030036_

Round 1

Reviewer 1 Report

The paper proposed a vibrator that generated vibration by utilizing the micro-vibration of an SMA wire and the mass-spring structure. It proves being advantageous over the piezoelectric actuator, electromagnetic actuators or electromagnetic actuators in different ways. The following issues are suggested to be addressed.

1. What is the reponse time for the phase transition from martensite phase to austenite phase?

2. The prototype vibrator seems not compact. How is its reliability?

3. What is the potential application senario of the proposed SMA vibrator?

The writting can be further polished.

Author Response

Dear Reviewer 1.

Sincerely,
Takashi Chujo

Reviewer 2 Report

 I am writing to provide my review comments on the paper titled "An application of micro-vibratory phenomena of an SMA wire to a novel vibrator" submitted to the Vibration Journal. I have thoroughly reviewed the manuscript and find it a valuable contribution to the haptic and vibrotactile displays field. Overall, the study presents a novel vibrator design utilizing shape memory alloy (SMA) wires and offers valuable insights into the control and evaluation of perceived vibrations. I commend the authors on their rigorous methodology and clear explanation of the experimental procedures.

Below, I have outlined my specific comments and suggestions for improving the manuscript:

Grammar:
The grammar and sentence structure are generally clear and understandable. However, attention to precise terminology and proofreading for any typographical errors or inconsistencies in punctuation would enhance the overall quality of the writing.

Abstract:
The abstract effectively summarizes the study's objectives, methods, and key findings. To improve it further, I suggest adding more specific details regarding the novelty of the research and its potential impact on the field. Additionally, ensure that the abstract accurately reflects the scope and significance of the study.

Scientific Background:
The scientific background provides a comprehensive overview of the motivation behind the development of the SMA-based vibrator. I recommend incorporating recent studies and advancements in the field to strengthen the scientific background. Additionally, discussing any limitations or challenges associated with existing actuators and highlighting how the proposed vibrator addresses those limitations would add depth to the section.

Methodology:
The methodology explanation provides a clear understanding of the experimental setup and procedures. However, providing more detailed information about the measurement techniques used to quantify the vibrations and the equipment employed would enhance the clarity of the methodology. Clear explanations regarding the participant selection process and details about their demographic characteristics are also important for ensuring the reproducibility of the study.

Providing Clear Discussion Based on the Results:
The discussion section offers insightful analysis and interpretation of the results. To further strengthen this section, I recommend comparing the findings with previous research in the field and discussing any discrepancies or similarities. If applicable, providing more detailed explanations for the observed trends and offering potential mechanisms or theoretical frameworks to support the interpretations would be valuable.
Conclusion:
The conclusion effectively summarizes the main findings of the study. To enhance it further, I suggest discussing the practical implications of the research and its potential impact on various applications. Additionally, providing recommendations for further research based on the study's outcomes and limitations would strengthen the conclusion.
In conclusion, the paper demonstrates a good understanding of the field and presents valuable findings. With the suggested improvements in precise terminology, incorporation of recent research, more detailed methodology explanations, and the inclusion of the mentioned comments, I believe this paper is suitable for publication in the Vibration journal.

Thank you for considering my review comments. Please feel free to reach out if you require any further clarification or assistance.

Author Response

Dear Reviewer 2.
Please see the attachment.

Sincerely,
Takashi Chujo

Reviewer 3 Report

It would be necessary to perform a more detailed study of the fast-flowing physical processes in the wire material under the influence of a pulsed changes in current

Minor corrections in the text and some improvements of the presentation style  are recommended

Author Response

Dear Reviewer 3.
Please see the attachment.

Sincerely,
Takashi Chujo

Reviewer 4 Report

Reviewers comments of manuscript, vibration-2450618, An application of micro-vibratory phenomena of an SMA wire to a novel vibrator

The manuscript deals with the study of an SMA wire as a vibrator device using the micro-vibratory phenomena, even importantly their potential vibration application results was presented in details in this MS. The quality of this MS probably validates its publication in Vibration, but it is not acceptable for publication in its present form. However, major revisions and corresponding clarify a couple of issues need to been done by the authors, as follow:

1. In introduction part, the authors should illustrate the specific reason or application advantage of SMA wires as vibrator device in comparison with other type methods. Meanwhile, the author also should refer to the latest progress and application of the micro-scale wires. Therefore, it is quite necessary to add the related works or statements, for different type microwires, and then elicit the research significance. To some extent, the SMA wires are conventional, and also lacks the novelty, so the authors should give the main reasons and consideration for the experimental design and explanation both in introduction and discussion parts.

2. According to this MS, the authors should give the composition of SMA wires, and present a model for determining micro-vibratory phenomena using SMA wires, and measure the samples of micro-vibratory of the vibrator device, but it is possibly hard to conclude or understand the further evidence of the micro-vibratory values increase depending on the frequency. Hereby, I strongly suggest the author should supplement the related statement to provide the directive evidence for the final explanation.

3. I have just listed a few but not all plotting and typing mistakes below:

Figure 1 & Figure 2:

I suggest the authors should reset the images, and uniform both the images too.

Figure 3:

I strongly suggest the authors should reduce the image proportion.

Page 6 & Figure 9 & Figure 12 & Figure 13:

There is some wrong arrangement for the (a) and (b), I suggest the authors should add them at the top left corner separately. And the same problem also exist in Figure 12 and Figure 13.

Conclusions part:

I suggest that the author should supplement the 2~3 items of conclusion sentences for quick information and reading.

References:

The journal abbreviation and typesetting in references should be done strictly according to the demand and rule of Vibration. i.e. Ref.[3], Ref.[5], Ref.[6], Ref.[10], etc.

In addition, the author should add some latest references for different type microwire applications into MS, such as https://doi.org/10.1002/adem.201700935, https://doi.org/10.1002/adem.201000204, etc.

English writing:

English writing of this MS could be improved further.

……

Author Response

Dear Reviewer 4.
Please see the attachment.

Sincerely,
Takashi Chujo

Round 2

Reviewer 4 Report

The authors have well answered my questions and comments in their revised MS, but I agree that this MS could be published on Vibration after English improvement and polishing again.

The authors have well answered my questions and comments in their revised MS, but I agree that this MS could be published on Vibration after English improvement and polishing again.